# mRNA-Driven Generation of Transgene-Free Neural Stem Cells from Human Urine-Derived Cells

**DOI:** 10.3390/cells8091043

**Published:** 2019-09-06

**Authors:** Phil Jun Kang, Daryeon Son, Tae Hee Ko, Wonjun Hong, Wonjin Yun, Jihoon Jang, Jong-Il Choi, Gwonhwa Song, Jangbo Lee, In Yong Kim, Seungkwon You

**Affiliations:** 1Institute of Animal Molecular Biotechnology, College of Life Sciences and Biotechnology, Korea University, Seoul 02841, Korea; 2Laboratory of Cell Function Regulation, Department of Biotechnology, College of Life Sciences and Biotechnology, Korea University, Seoul 02841, Korea; 3Division of Cardiology, Department of Internal Medicine, Korea University College of Medicine and Korea University Medical Center, Seoul 02841, Korea; 4Cardiovascular Research Institute, Korea University, Seoul 02841, South Korea; 5Department of Neurosurgery, College of Medicine, Korea University, Seoul 02841, Korea

**Keywords:** induced neural stem cells (iNSCs), self-replicative mRNA, direct conversion, reprogramming, small molecules, neurological diseases

## Abstract

Human neural stem cells (NSCs) hold enormous promise for neurological disorders, typically requiring their expandable and differentiable properties for regeneration of damaged neural tissues. Despite the therapeutic potential of induced NSCs (iNSCs), a major challenge for clinical feasibility is the presence of integrated transgenes in the host genome, contributing to the risk for undesired genotoxicity and tumorigenesis. Here, we describe the advanced transgene-free generation of iNSCs from human urine-derived cells (HUCs) by combining a cocktail of defined small molecules with self-replicable mRNA delivery. The established iNSCs were completely transgene-free in their cytosol and genome and further resembled human embryonic stem cell-derived NSCs in the morphology, biological characteristics, global gene expression, and potential to differentiate into functional neurons, astrocytes, and oligodendrocytes. Moreover, iNSC colonies were observed within eight days under optimized conditions, and no teratomas formed in vivo, implying the absence of pluripotent cells. This study proposes an approach to generate transplantable iNSCs that can be broadly applied for neurological disorders in a safe, efficient, and patient-specific manner.

## 1. Introduction

Most neurological disorders lead to irreversible damage to the central nervous system (CNS) upon disease progression due to the limited potential of CNS repair via endogenous neurogenesis. This medical dilemma has been lasted before the discovery of neural stem cells (NSCs) in the 1960s, offered considerable promise for CNS regeneration in patients [1]. Since then, several studies have focused on expansion of human NSCs in vitro and evidence of neurogenesis in several regions of the mammalian CNS [2,3,4,5]. However, the scarcity and poor accessibility of primary NSC populations from adult neurogenesis hinder their preclinical and clinical applications [6,7]. In this context, the generation of NSC populations from pluripotent stem cells (PSCs), including embryonic stem cells (ESCs) and induced pluripotent stem cells (iPSCs), has been spotlighted as a promising alternative. Especially, because of the major issues facing immune rejection and ethical concerns for transplantation of ESC-derived NSCs in clinical trials, the advent of patient-specific iPSCs obtained from somatic cells has been hailed as a revolutionary source for personalized medicine [8]. The generation of iPSCs by forced expression of a defined set of transcription factors, OCT4, SOX2, KLF4, and c-MYC (also known as Yamanaka factors) is a fairly straightforward and robust technology, which has been explored for a better understanding of human development and diseases processes, discovering therapeutics [9]. Meanwhile, for their clinical trial transparency, the re-differentiated cells from iPSCs must be addressed in feasibility in precise control of PSCs differentiation and formation of teratomas in vivo by undifferentiated population from PSCs. For bypassing a pluripotent stage, direct conversion of cell fates employing the cell-activation and signaling-directed (CASD)-induced lineage transition, mediated by Yamanaka factors and lineage-specifying cocktail of small molecules, has enable to produce a wide range of cell types, including neurons [10], neural progenitors [11], NSCs [12], cardiomyocytes [13], and hepatocyte-like cells [14]. Despite these impressive exhibitions of cell fate conversions, the viral vector-mediated gene delivery system lead to chromosomal integration of exogenes in host genome, restricting their access to clinical trials. Related studies have shown encouraging results in avoiding the chromosomal integration by employing non-integrative delivery systems, including episomal plasmids, adenovirus, and Sendai virus [15,16]. However, these methods require a time- and cost-consuming process to permanently erase the residual transgenes from host cells for preventing the risks caused by uncontrolled exogenes reactivation [17,18]. Another study using synthetic mRNA encoding the reprogramming factors has showed the generation of integration-free iPSCs without additional transgene deletion process [19], nonetheless, the use of extremely short-lived mRNA species required to repeat the transfection during conversion process, which could trigger to cellular stress responses, thus leading to poor reproducibility. In contrast, self-replicative mRNA delivery system employing Venezuelan equine encephalitis (VEE) replicon have showed the generation of iPSCs with high efficiency in a single transfection, thereby contributing to fine reproducibility in mRNA-based reprogramming [20]. Therefore, a self-replicable mRNA delivery system would be an excellent candidate for clinical applicable CASD-based lineage transition.

On the other hand, efforts in minimizing the genetic manipulation in reprogramming have been extensively studied due to the safety issues for its clinical applications. One of the most promising solutions is a chemical approach using small molecules, enabling to regulate signal pathways and epigenetic status, thus improving quality and efficiency of cell fate conversion [21]. Although completely replacing all of reprogramming factors with small molecules still remain an open question, at least in human cells, recent studies have widely employed several small molecules to reduce the requirement for exogenous factors [22,23,24,25,26]. Based on these recent advances, a chemical approach using small molecules could be a promising synergistic strategy for guiding cell fate determination.

In this study, we focused our efforts on developing a strategy for generation of human induced NSCs (iNSCs) in terms of: (1) accessibility of source cells for reprogramming, (2) simplicity in conversion processing, (3) accuracy of cell fate manipulation, (4) verification of the integration-free method, and (5) high reproducibility and efficiency. We employed self-replicative mRNA replicon system for controllable expression of OCT4, KLF4, SOX2, and GLIS1 (OKSG), which has been previously reported to reprogram somatic cells into iPSCs without tumorigenic c-MYC [20]. This system was used together with combination of small molecules essential for efficient completing the direct reprogramming of human urine-derived cells (HUCs) into iNSCs. The iNSC colonies were observed within eight days under optimized conditions without pluripotency. The established iNSCs were completely transgene-free in their cytosol and genome and resembled human embryonic stem cell-derived NSCs in the morphology, biological characteristics, and global gene expression, as well as potential to differentiate into functional neurons, astrocytes, and oligodendrocytes. This study proposes an approach to generate transplantable iNSCs that can be broadly applied for neurological disorders in a safe, efficient, and patient-specific manner.

## 2. Materials and Methods

### 2.1. Generation of iNSCs from HUCs

This study was approved by the Institutional Review Board at the Korea University (IRB approval number; 1040548-KU-IRB-18-62-A-2). We used the urine samples of three males and one female in this study. The analysis from sample of one male (age 33) was presented as a representative for the establishment of iNSCs in the main manuscript. Primary HUCs were cultured in urine growth medium (DMEM:REGM (Lonza, Basel, Switzerland) (1:1), 10% FBS (Fetal bovine serum) (Hyclone, Logan, UT, USA), 1× penicillin/streptomycin, 1× l-glutamine, 1× non-essential amino acids, 5 ng/mL FGF2 (Peprotech, Rocky Hill, NJ, USA) and 5 ng/mL EGF (Peprotech)). To generate iNSCs from HUCs, 1 × 10^6^ cells of HUCs were transfected with synthetic mRNAs, including OCT4, SOX2, KLF4, GLIS1, and B18R, by electroporation (MP-100, Thermo Fisher Scientific, Waltham, MA, USA). Synthetic mRNAs were collected by in vitro transcription using DNA plasmid templates as provided by Addgene (ID58974 and 58978). At day two after transfection, HUCs were re-seeded on Matrigel-coated plates with B18R protein. For generating iNSCs, HUCs were cultured in the reprogramming medium consisting of DMEM/F12: Neurobasal (Thermo Fisher Scientific) (1:1), 1× N2 (Thermo Fisher Scientific), 1× B27 (Thermo Fisher Scientific), 1x penicillin/streptomycin, 1× l-glutamine, 1× non-essential amino acids, 10 ng/mL recombinant human LIF (MilliporeSigma, Burlington, MA, USA), 2 μM SB431542 (Tocris Bioscience, Missouri, UK) and 3 μM CHIR99021 (Tocris Bioscience) (LSC medium) with small molecules (0.5 μM Purmorphamine (Tocris Bioscience), 10 μM Forskolin (Tocris Bioscience), 64 μg/mL Vitamin C (MilliporeSigma), and 100 μM Sodium butyrate (Tocris Bioscience)). iNSC colonies were observed within eight days after induction. At 12 days after induction, neuroepithelial-like iNSC colonies were picked up and transferred onto Matrigel-coated plates in LSC medium. iNSCs expanded in LSC medium and subcultured using Accutase (STEM CELL Technologies, Vancouver, BC, Canada). Established iNSCs (passage 5–10) are used for characterization in this study. 

### 2.2. In Vitro Differentiation of iNSCs

For spontaneous differentiation, iNSCs were seeded on PLO/Laminin coated plates and cultured in a differentiation medium consisting of DMEM/F12, 1× N2, 1× B27, 300 ng/mL cAMP (Tocris Bioscience), 64 μg/mL Vitamin C, 10 ng/mL BDNF (Peprotech), and 10 ng/mL GDNF (Peprotech) for two weeks. For astrocyte differentiation, iNSCs were seeded on PLO/Laminin coated plates and cultured in a differentiation medium consisting of DMEM/F12, 1× N2, 1× B27, 20 ng/mL CNTF (Peprotech), and 10 ng/mL BMP4 (Peprotech) for 1–2 weeks. For GABA neuron differentiation, iNSCs were seeded on PLO/Laminin coated plates and cultured in a differentiation medium consisting of DMEM/F12, 1× N2, 0.5× B27, and 300 ng/mL cAMP (Tocris Bioscience) for three weeks. For TH neuron differentiation, iNSCs were seeded on PLO/Laminin coated plates and cultured in a ventral midbrain-specified medium consisting of DMEM/F12, 1× N2, 200 ng/mL sonic hedgehog (SHH) (Peprotech), 100 ng/mL FGF8b (Peprotech), and 50 μg/mL Vitamin C for one week and then cultured in a differentiation medium consisting of DMEM/F12, 1× N2, 20 ng/mL BDNF, 10 ng/mL GDNF, and 50 μg/mL Vitamin C for two weeks. For HB9 neuron differentiation, iNSCs were seeded on PLO/Laminin coated plates and cultured in a posterior hindbrain- and spinal cord-specified medium consisting of DMEM/F12, 1× N2, 1× B27, 20 μg/mL Insulin (MilliporeSigma), 10 ng/mL FGF2, 10 ng/mL EGF, 1 μM Retinoic acid, and 1 μg/mL SHH for one week and, after removal of FGF2 and EGF, cultured for one more week. Finally, cells were differentiated in DMEM/F12, 1× N2, 1× B27, 20 ng/mL BDNF, 20 ng/mL GDNF, and 50 ng/mL SHH for two weeks. For oligodendrocyte differentiation, iNSCs were seeded on PLO/Laminin coated plates and cultured in a differentiation medium consisting of DMEM/F12, 1× N2, 1× B27, 25 μg/mL Insulin, 0.1 μM Retinoic acid, and 1 μM SAG (Tocris Bioscience) for OLIG2 progenitor. Afterwards, further differentiation in DMEM/F12, 1× N2, 1× B27, 25 μg/mL Insulin, 10 ng/mL PDGFaa (Peprotech), 10 ng/mL IGF-1 (Peprotech), 5 ng/mL HGF (Peprotech), 10 ng/mL NT-3 (Peprotech), 60 ng/mL T3 (MilliporeSigma), 100 ng/mL Biotin (MilliporeSigma), and 1 μM cAMP was performed for oligodendrocyte progenitor, and then maturated in DMEM/F12, 1× N2, 1× B27, 25 μg/mL Insulin, 10 mM HEPES (MilliporeSigma), 60 ng/mL T3, 100 ng/mL Biotin, 1 μM cAMP, and 20 μg/mL Vitamin C for oligodendrocyte.

### 2.3. In Vivo Differentiation of iNSCs

Animal study was approved by the Institutional Animal Care & Use Committee at the Korea University (IACUC approval number; KUIACUC-2018-12). iNSC mass (1 × 10^5^ cells) in artificial cerebrospinal fluid (MilliporeSigma) were injected into brain (ML = 2mm, AP = 1mm and DV = −2.5mm) of BALB/c Nude mice through a 26G syringe (Hamilton, Reno, NV, USA). After eight weeks post-transplantation, mice were perfused with 4% paraformaldehyde (PFA), and their brains were harvested. The isolated brains were fixed in 4% PFA overnight and transferred into 30% sucrose in PBS at 4 °C for 48 h. Brains were embedded in OCT compound and cut into 16–18 μm sections. For antigen recovery, the tissue sections were boiled for 5 min in boiling citrate buffer (41 mL of 0.1 M Na Citrate, 9 mL of 0.1 M Citric acid, and 450 mL of distilled water) and then incubated in 1% SDS (sodium dodecyl sulfate) in PBS for 5 min. The tissue samples were immersed in PBS containing 0.3% Triton X-100 and 2% donkey serum for 40 min at room temperature and then incubated with primary antibodies in PBS containing 2% donkey serum overnight at 4 °C. Afterwards, the samples were incubated with secondary antibodies for 1 h at room temperature and then nuclei were stained with DAPI for 5 min at room temperature. Immunofluorescence was visualized under Confocal Laser Scanning Microscope (LSM 700, ZEISS, Oberkochen, Germany).

### 2.4. RT-PCR and qRT-PCR

Total RNA was isolated from cells by TRIzol (Invitrogen, Carlsbad, CA, USA), and cDNA was synthesized using AccuPower^®^ RT-PreMix (Bioneer, Daejeon, South Korea) with oligo-dT-18 primer (Bioneer). Target gene fragments were amplified by specific primers. Beta-actin was used as a housekeeping gene (normalization control). qRT-PCR reactions were performed with iQ SYBR Green Supermix (Bio-Rad, Hercules, CA, USA).

### 2.5. Immunocytochemistry

Cells were fixed in 4% PFA for 20 min at room temperature. Permeabilization and blocking was proceeded in PBS containing 0.3% Triton X-100 and 5% donkey serum in PBS for 20 min at room temperature and then incubated with primary antibodies in PBS containing 5% donkey serum at 4 °C overnight. Cells were incubated with secondary antibodies for 1 h at room temperature and then nuclei were stained with DAPI for 5 min at room temperature. Specimen was rinsed twice with PBS at each step. Images of immunofluorescence were visualized under a fluorescence microscope (IX71, OLYMPUS Tokyo, Japan).

### 2.6. Karyotype Analysis

Karyotyping were conducted by GTG banding by Samkwang Medical Laboratories (Seoul, South Korea).

### 2.7. Teratoma Formation

Animal study was approved by the Institutional Animal Care & Use Committee at the Korea University (IACUC approval number; KUIACUC-2018-0018). A million of ESCs or iNSCs was harvested, resuspended in 100 μL cell expansion medium, and mixed with an equal volume of Matrigel stock solution. Cells were injected into dorsal flank of BALB/c Nude mice. Afterwards, teratoma formation was monitored over four months.

### 2.8. RNA-Sequencing Experiment

For RNA-sequencing, rRNA was removed from total RNAs (each 5 ug) by using Ribo-Zero Magnetic kit (epicentre, Inc., Madison, WI, USA). Genomic libraries were constructed using SENSE Total RNA-Seq Library Prep Kit (Lexogen, Inc., Vienna, Austria) according to the manufacturer’s protocol. The starter/stopper heterodimers, containing Illumina-compatible linker sequences, were randomly hybridized to remained RNA, initiating library production. The starter was extended to the next hybridized heterodimer by a single-tube reverse transcription and ligation reaction, followed by the stopper ligated with newly-synthesized cDNA inserts. The resulting library was amplified, following second strand synthesis to release the library from beads, thus introducing barcodes. As paired-end 100 sequencing, the sequencing was performed by HiSeq 2500 (Illumina, Inc., San Diego, CA, USA). The FPKM (fragments per kilobase of exon per million fragments) was used for determining the expression levels of gene regions.

### 2.9. Whole Cell Patch Clamp Recordings

Patch clamp experiments were performed using Axopatch 200B amplifier (Axon Instrument) at room temperature (23 ± 1 °C). The iNSCs were placed in a chamber mounted on to inverted microscope and continuously superfused with artificial cerebrospinal fluid (aCSF) solution containing NaCl 126, KCl 3, MgSO_4_ 2, CaCl_2_ 2, NaH_2_PO_4_ 1.25, glucose 10, and HEPES 10 in mM (pH 7.4). The patch pipette solution contained KCl 140, EGTA 5, glucose 5, HEPES 5, Mg-ATP 5, and MgCl_2_ 1 in mM (pH 7.2). Patch pipettes were pulled from thin-walled borosilicate capillaries (Clark Electromedical Instruments, Pangbourne, UK) using a P-97 Flaming/Brown Micropipette Puller (Sutter Instrument Company, Novato, CA, USA). Glass microelectrodes were pulled, yielding a tip resistance of 4–6 MΩ when filled with pipette solution. All the recordings were carried out at every 5 min after making whole-cell configuration to allow cells. The voltage and current signals were filtered at 10 kHz, 4-pole Bessel type low-pass filter and sampled at a rate of 25 kHz. Cell capacitance (pF) and access resistance (MΩ) were automatically calculated and used to compensate for capacitive current and to normalize ion current (pA/pF). Data acquisition and analysis were performed using digitizers (DigiData 1550B) and analysis software pClamp 10.7 (Molecular Devices). Tetrodotoxin (TTX) was used to block voltage-gated Na+ channels and tetraethylammonium chloride (TEA-Cl) was used to block K^+^ channels.

### 2.10. Statistical Analysis

All values are expressed as mean ± SD. Data comparisons were made using unpaired, two-tailed Student’s *t* test. A p value less than 0.05 was considered statistically significant. * *p* < 0.05, ** *p* < 0.01. All experiments were performed independently at least three times.

### 2.11. Accession Numbers

RNA-sequencing data have been submitted and can be accessed by the Gene Expression Omnibus (GEO) accession number GSE119669.

## 3. Results

### 3.1. Optimized-Conditioning by Small Molecules for Generating iNSCs from HUCs

We first evaluated the transfection efficiency of VEE-GFP-RNA replicon via electroporation into HUCs. At two days post-transfection, 72.2% of cells were GFP+ (Figure 1A,B). For generation of integration-free iNSCs, a self-replicating VEE-RNA encoding the reprogramming factors OKSG served as a critical tool in this study. Based on previous reports [15,27], we attempted to generate iNSCs on the RNA replicon system, followed by culture in chemically defined medium containing leukemia inhibitory factor (LIF), SB431542, and CHIR99021 (LSC medium) for 15 days (Figure 1C); however, none or few neuroepithelial colonies were observed (Figure 1D). This suggests that the condition used for Sendai virus-mediated generation of iNSCs [15] are insufficient for this RNA-based system. To explore the molecular cues governing the cell fate, we employed small molecules Purmorphamine (P), Forskolin (F), Vitamin C (V), and Sodium butyrate (N) which were related to reprogramming and neural differentiation, alone, or in combination (Figure 1C) [22,25,26,28,29,30,31,32,33,34]. As a result, neuroepithelial colonies were observed in cultures exposed to P, F, V, and N alone or in combination (Figure 1D). The number of colonies was significantly increased upon exposure to PFVN (Figure 1D). These findings were supported by comparing colony formation efficiency of SOX1+ and PLZF+ cells in individual removal of P, F, V, and N (Figure 1E). In this result, we next evaluated exposure duration of B18R protein which is critical regulator of exogenous mRNA expression. Previous report suggested that treatment of B18R protein is required during whole reprogramming process for iPSC generation using RNA replicon system [20], however, other reports implied only a short-term period of exogene expression is required for iNSC generation using Sendai virus [15]. Therefore, we first transfected GFP-encoded VEE-RNA into foreskin fibroblasts for investigating relationship between B18R protein treatment and exogene expression. As expected, withdrawing of B18R proteins led to rapid decrease of GFP expression in both terms of efficiency and intensity, and it eventually dissipated within seven days (Appendix A). Next, we treated B18R protein at various time points during iNSC induction. Interestingly, iNSC colonies were successfully collected through exposure to B18R protein only during the growth period (D-3 to D0); B18R protein was not required during the reprogramming period (D0 to D12) (Figure 1F). This suggests that iNSC allowed very restricted dependency on exogenous expression for induction. Our protocol clearly showed a gradual increase of PLZF and endogenous SOX2 expression, whereas the expression of pluripotent genes was restricted over time (Figure 1G,H). In addition, to assess the effects of PFVN treatment in combination with either normoxic or hypoxic conditions which conventionally enhanced reprogramming efficiency via decrease in ROS damage, conversion in glycolytic metabolism, and HIF induction [35], we induced HUCs to iNSCs under normoxia or hypoxia conditions. As expected, hypoxic exposure resulted in more than two-fold increase in SOX1+/PLZF+ colony formation compared to normoxic condition (Figure 1I). In this optimized condition, iNSCs were produced within eight days (Figure 2A–E), while iPSCs are generated in 25 days using a similar RNA-based system (data not shown) [20]. We established iNSC lines from HUCs of four healthy donors (three males and one female) under optimized conditions. The iNSCs expressed NSC markers, including SOX1, SOX2, NESTIN, PAX6, and PLZF, comparably with H9-ESC derived NSCs (H9-NSCs), as a positive control (Figure 2F–I, and Appendix A). We confirmed the identity of original HUCs and iNSCs by short tandem repeat (STR) analysis (Appendix A).

### 3.2. Characterization of iNSCs from HUCs by Synthetic mRNA with Small Molecules

The iNSCs established from HUCs formed neuroepithelial-like colony within eight days and stably survived in conventional primitive NSC medium described previously (Figure 2A–E) [15,27]. The iNSCs expressed typical NSC markers, such as NESTIN, SOX2, SSEA1, PLZF, and PAX6 (Figure 2F–I), which were similar to those observed in H9-NSCs (Figure 2J), with high proliferative potential by expression of Ki67 (Figure 2K). The iNSCs, however, did not express the pluripotency markers OCT4 and NANOG during in vitro expansion (Figure 2L,M). To determine whether iNSCs possess a regional identity, we assessed region-specific markers via RT-PCR analysis. Although both H9-NSC and iNSCs share expression of fore-/hind-brain and spinal cord markers, such as FOXG1, OTX1, GBX2, HOXB2, and HOXB4, the expression of OTX2 and EN1 as fore-/mid-brain markers was observed only in iNSCs (Figure 3A). Interestingly, region-specific gene expression profiling analyzed by RNA sequencing indicates that posterior specification and dorsalization of iNSCs were enriched, which are similar to those of H9-NSCs, that could be explained as continuously activated Wnt signaling mediated by CHIR99021 supplemented to the NSC medium (Figure 3B,C). To confirm the plasticity of regional identity of iNSCs, we treated purmorphamine or FGF8b to induce ventral or midbrain properties of iNSCs, respectively. As a result, iNSCs were able to ventralize by purmorphamine-induced Shh signaling via increased markers expression, such as OLIG2, NKX2.2, and NKX6.1, with decreased expression of PAX3 and PAX6 (Figure 3D–I). Only a small portion of iNSCs, however, expressed EN1 and OTX2 in response to FGF8b, suggesting iNSCs were patterned caudalized regionality (Figure 3J–M).

We compared the global gene expression patterns of iNSCs with those of parental urine cells, as well as H9-NSCs by RNA sequencing. Hierarchical clustering and pairwise scatter plots showed the genome-wide conversion of HUCs into iNSCs with a high degree of similarity between H9-NSCs and iNSCs (Figure 4A–C). Compared to parental HUCs, the gene ontology categories, which showed significantly enriched in iNSCs, were gene sets related to nervous system development and neurogenesis; in contrast, cell adhesion-related genes were highly downregulated in iNSCs (Figure 4D–H). In addition, the expression profiles of non-translated lncRNA also showed a high degree of similarity between H9-NSC and iNSCs (Appendix A). These results suggest that HUCs were converted into NSC fate during the reprogramming process.

### 3.3. In Vitro/Vivo Differentiation Potential of iNSCs and Electrophysiological Assessment

Clinical feasibility of established iNSCs relies on their differentiation capability into functional subtypes of neural cells including neurons, astrocytes, and oligodendrocytes. The established iNSC lines transferred to the specified medium compositions for inducing differentiation into neural subtypes as previously reported [27]. After differentiation of the established iNSC lines, we observed TUJ1+ and MAP2+ (early and mature neuronal markers) neurons with markers expression of GABA+ (GABA neuron), TH+ (dopaminergic neuron), and HB9+ (motor neuron) (Figure 5A–D), as well as synaptophysin, a synaptic vesicle glycoprotein participated in synaptic transmission, in the differentiated neurons (Figure 5E). In addition, action potential (AP) in iNSC-derived neurons (three weeks for differentiation) was shown in response to depolarizing current injection and resting membrane potential (RMP) was measured at −58.10 mV ± 7.62 by whole-cell patch clamp analysis (Figure 5F,G). Microelectrode array (MEA) system showed spontaneous electronic spikes induced by iNSC-derived neurons (Figure 5I). The voltage-clamp for sodium and potassium channel recordings were performed before and after application of 5 μM TTX and 10 mM TEA, which showed the sodium and potassium currents were significantly blocked by application of TTX (red, −20 to 0 mV) and TEA (blue, −20 to +40 mV) in iNSC-derived neurons (Figure 5J–M).

Gliogenesis properties of iNSCs were observed under astrocyte differentiation condition with S100β+/GFAP+ (astrocyte markers) (Figure 5O) expression and oligodendrocyte differentiation condition with OLIG2+, PDGFRα+, and O4+ oligodendrocyte progenitors, as well as MBP+ oligodendrocyte (Figure 5P–R). The multipotency of iNSCs was repeatedly confirmed in the iNSC lines derived from three different donors (Appendix A). In order to examine the differentiation potential of iNSCs in vivo, we generated the iNSCs infected with lentiviral vector encoding GFP (Figure 6A,B) and transplanted into the mice brain (Figure 6C). At eight weeks post-transplantation, the GFP-positive cells were colocalized with cells positive for MAP2, GFAP, and MBP as neuron, astrocyte, and oligodendrocyte markers, respectively (Figure 6D–F). Thus, these results demonstrate the clinical potential of iNSCs through their differentiation capacity into functional neurons, astrocytes, and oligodendrocytes in vitro/in vivo.

### 3.4. Biosafety Evaluations of iNSCs

To assess the biological safety of iNSCs, we first evaluated chromosomal stability by karyotyping analysis. The iNSCs established from four healthy donors preserved a normal karyotype (Figure 7A and Appendix A) and no significant differences in cell doubling time with normal karyotype between passages 8 and 20 (Figure 7B,C). Next, we confirmed existence of the residual VEE-RNA species in the cytosol of iNSCs or integration to their genomic DNA during iNSCs generation. We traced VEE-RNA species from iNSCs by PCR when compared to VEE-OKSiG RNA-transfected HUCs at 48 h post-transfection, serving as a positive control. As shown in Figure 7D, no VEE expression was detected in the RNA pool of HUC derived-iNSCs and their genomic DNA (Figure 7D). Additionally, the injected iNSCs into back trunk of immunodeficient nude mice did not form teratomas over four months, comparable to the ESC sites (Figure 7F,G), and when transplanted into brain of nude mice, did not induce aberrant tissue morphology over six months, implying the absence of tumorigenic potential of iNSCs (Figure 7H). These findings clearly indicate that the established iNSCs are transgene-free with lack of tumorigenicity.

## 4. Discussion

The generation of integration-free iNSCs from human HUCs was successfully achieved by controlled replication of self-replicative mRNAs encoding reprogramming factors with combination of small-molecules under hypoxic condition. The present protocol employed a simple, safe, and efficient means to generate iNSCs by a single transfection of self-replicating polycistronic mRNAs, with controlling reprogramming factors expression using B18R protein. The present study also provides the optimized condition to reprogram somatic cells to NSC fate commitment, bypassing the pluripotent stem cell stage. Lu et al. have reported that, using a temporal expression of Sendai virus, the directly converted NSCs from fibroblasts showed a high neuronal potential and self-renewing capacity with elimination of fibroblast growth factor 2, which is absolutely essential for iPSC induction [15]. Based on their work, we attempted to generate iNSCs using self-replicating mRNA followed by culture in the induction condition as previously described [15]; however, it was unsuccessful, which might be due to an insufficient expression level of the reprograming factors compared to the Sendai virus-mediated expression (data not shown). To overcome this, we employed small molecules which enable to compensate for the reprogramming factors, including purmorphamine (sonic hedgehog (SHH) agonist), forskolin (cyclic adenosine monophosphate (cAMP) activator), sodium butyrate (histone deacetylase inhibitor), and ascorbic acid (antioxidant and epigenetic modifier) (Figure 8). These molecules have been widely used to increase reprogramming efficiencies or even manipulate neural development by combining with other compounds [22,25,26,28,29,30,31,32,33,34]. For example, the SHH signaling plays important roles in ventral CNS development, as well as contributes to differentiating NSCs into motor neurons and oligodendrocytes [36]. Related studies showed the generation of NSCs from mouse embryonic fibroblasts using chemical combinations including SHH signaling agonists [25,26]. Meanwhile, cAMP analogs/activators have usually been used for neuronal differentiation and glial maturation in vitro [15,27]. In adult brains of mice and humans, cAMP mediates the potential activities of cAMP response element binding protein (CREB) in multiple developmental steps of neurogenesis, thus acting as a second messenger for regulating neuro-specific signaling pathway [32,33]. In the field of cell reprogramming, forskolin is known as a cAMP activator, possibly enabling the replacement of Yamanaka factors for reprogramming of mouse cells [22]. Sodium butyrate is an epigenetic modulator that induces histone acetylation to reconstitute the original epigenetic memory of cells during reprogramming [31]. Although ascorbic acid is a well-known antioxidant, which can reduce the risk of genetic mutation by regulating intracellular levels of reactive oxygen species, it has also been reported as an epigenetic modifier that increases histone and DNA demethylation [28,29]. Another study reported that ascorbic acid induces the differentiation of ESCs into neurons by modulating expression of gene sets involved in cell adhesion and development [34]. Furthermore, a previous study described the generation of NSCs by small molecules under mild hypoxic condition (2.5–5.0% O_2_) [35]. Based on this, we employed hypoxic conditioning to enhance self-renewal ability of NSCs by mimicking their natural niche environment [37,38]. Overall, our strategy allowed the collection of NSC colonies within eight days, thereby not only permitting rapid direct conversion bypassing the pluripotent stage but also improving the reprogramming efficiency and quality.

## 5. Conclusions

Despite the enormous promise of stem cells for patient-specific regenerative medicine therapies, clinicians have continually questioned their value versus the risks of residual transgenes. For example, Nakagawa et al. have reported that retroviral OSKM-driven iPSCs lead to cancer development in the derived chimeric mice due to lasting or accidentally rebound expression of transgene c-MYC integrated in the host genome [39,40]. Another study showed that the expression of residual transgenes affects the transcriptional program and epigenetic signatures of reprogrammed cells even after the conversion processing is completed [41]. Alternatively, one of the non-DNA-based approaches is transfection of reprogramming mRNAs synthesized in vitro [19]. However, due to the short life-time of mRNAs, repetitive transfection has inevitably required over the reprogramming period. Here, the VEE-RNA-based transfection allowed a time-controllable expression of transgene RNAs by B18R protein treatment, withdrawal of B18R protein leads complete elimination of VEE-RNAs in the target cells [20]. Taken together, these advantages propose high compatibility to establish patient-specific disease modeling and further therapeutic transplantation for various neurological disorders. Nevertheless, we expect that further study on employing newly screened epigenetic modifiers could improve our protocol in a qualitative and quantitative manner and further investigation for clinical feasibility of the established iNSCs (e.g., mutagenic property, therapeutic functionality in various animal models, etc.) should be performed according to strict criteria.

## Figures and Tables

**Figure 1 cells-08-01043-f001:**
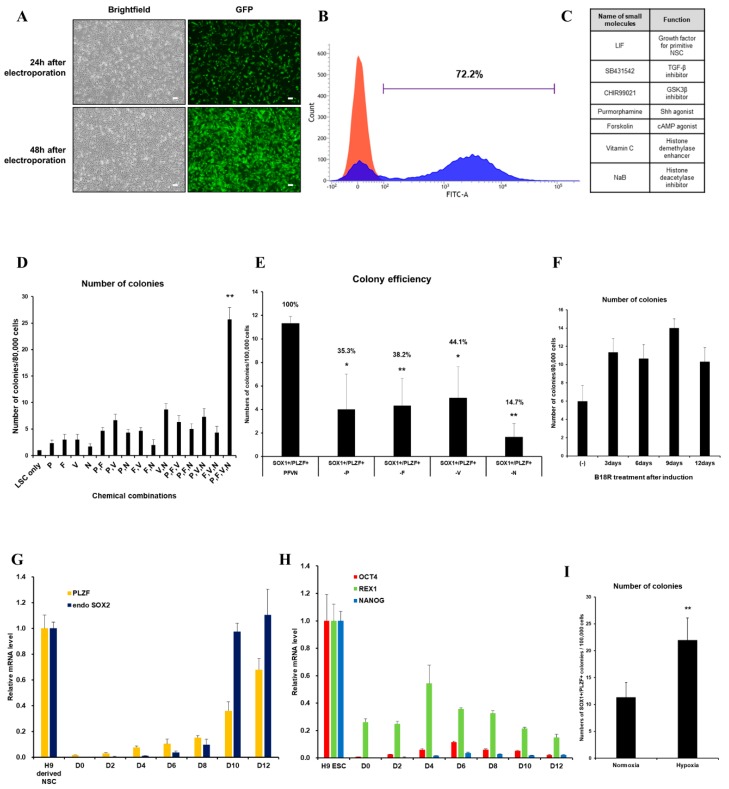
Optimized-conditioning by small molecules for generating induced neural stem cells (iNSCs) from human urine-derived cells (HUCs). (**A**) GFP (Green fluorescence protein) expression in HUCs at 24 and 48 h after electroporation. (**B**) FACS (Fluorescence-activated cell sorting) analysis showed high transfection efficiency to HUCs using synthetic mRNA encoding GFP. (**C**) Small molecules for generating iNSCs from HUCs. (**D**) Treatment with combinations of small molecules (P: purmorphamine, F: forskolin, V: ascorbic acid, N: sodium butyrate) increases efficiency for generating iNSC colonies. (**E**) Exclusion of a small molecule decreases the efficiency for generating iNSC colonies. (**F**) Efficiency for generating iNSC with or without B18R protein during induction. (**G**) qRT-PCR analysis of PLZF and endogenous SOX2 expression during iNSC generation. (**H**) qRT-PCR analysis of OCT4, REX1, and NANOG expression during iNSC generation. (**I**) Hypoxic condition increases the efficiency for generating iNSC colonies. Error bars represent standard deviation. * *p* ≤ 0.05; ** *p* ≤ 0.01.

**Figure 2 cells-08-01043-f002:**
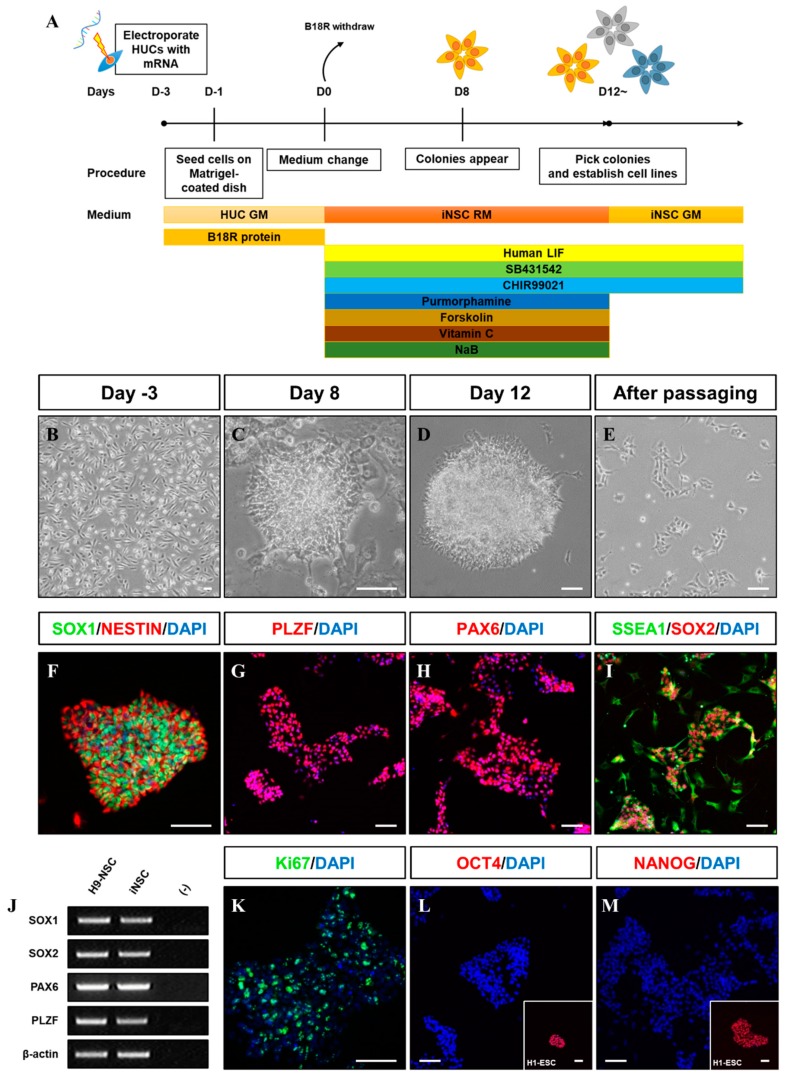
Generation of iNSCs from HUCs by self-replicating mRNAs with small molecules. (**A**) Schematic of the time course of the process used to directly convert HUCs into iNSCs. (**B**) Morphology of HUCs after transfection of self-replicating mRNAs. (**C**,**D**) Morphology of iNSC colonies exhibit neuroepithelial colony formation on day eight and 12. (**E**) Morphology of iNSCs after passaging using Accutase solution. (**F**–**I**) NSC markers (SOX1, NESTIN, PLZF, PAX6, SSEA1, and SOX2) expression in iNSC colonies, as determined by immunofluorescence analysis. (**J**) NSC markers are expressed in iNSCs similar with H9-NSCs, as demonstrated by RT-PCR. (**K**) Ki67 staining in iNSCs, as determined by immunofluorescence analysis. (**L**,**M**) OCT4 and NANOG expression in iNSCs and H9-ESCs, as determined by immunofluorescence analysis. Scale bars, 200 μm.

**Figure 3 cells-08-01043-f003:**
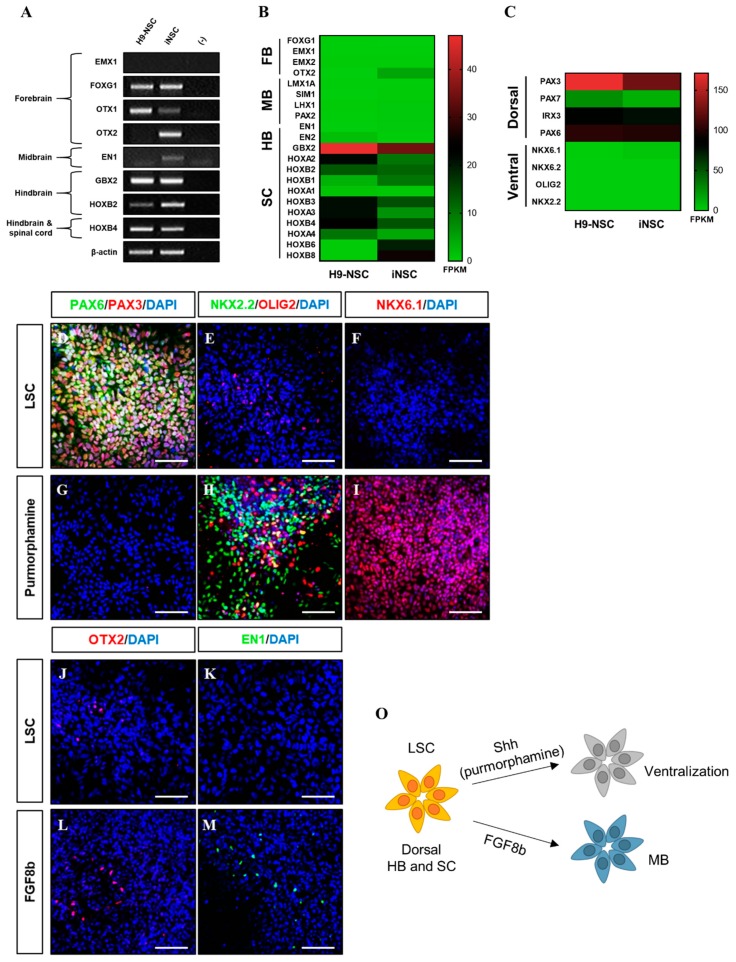
Developmental region-specific patterning of iNSCs. (**A**) RT-PCR analysis of forebrain, midbrain, hindbrain, and spinal cord markers in iNSCs and H9-NSCs. (**B**) Analysis of central nervous system (CNS) region-specific maker expression in H9-NSCs and iNSCs by RNA sequencing. FB, forebrain; MB, midbrain; HB, hindbrain; and SC, spinal cord. (**C**) Identification of dorsal and ventral-specific marker expression in H9-NSCs and iNSCs by RNA sequencing. (**D**–**I**) Purmorphamine treatment induces ventral markers (NKX2.2, OLIG2, and NKX6.1) and decreases dorsal markers (PAX3 and PAX6) of iNSCs, as determined by immunofluorescence analysis. (**J**–**M**) Forebrain marker (OTX2) and midbrain marker (EN1) expression in iNSCs under LSC culture medium or FGF8b treatment, as determined by immunofluorescence analysis. (**O**) Schematic diagram of region-specific patterning of iNSCs. Scale bars, 200 μm.

**Figure 4 cells-08-01043-f004:**
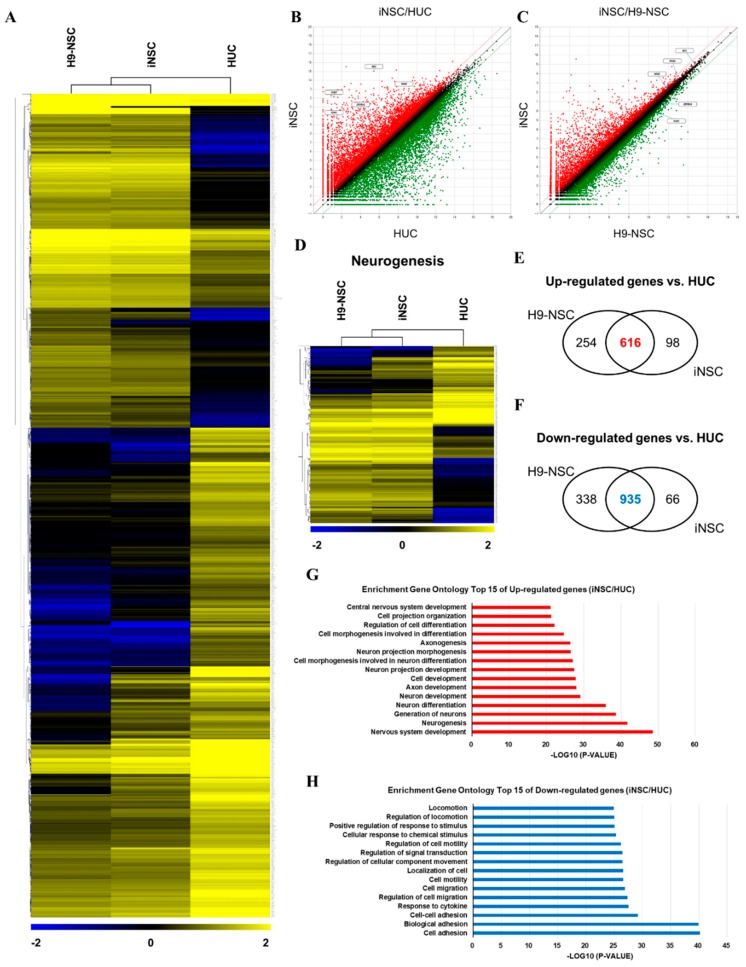
RNA sequencing analysis of the global gene expression profile of iNSCs. (**A**) Hierarchical clustering analysis of global gene expression in H9-NSCs, iNSCs, and HUCs. Two thousand two hundred and thirty-seven genes are selected with significant p-value (*p* < 0.05) and fold change (FC > 2). (**B**,**C**) Scatter plots of gene expression in iNSC vs. H9-NSCs or HUCs. (**D**) Hierarchical clustering analysis of gene expression related to neurogenesis in H9-NSCs, iNSCs, and HUCs. Two hundred ninety-nine genes are selected with significant p-value (*p* < 0.05) and fold change (FC > 2). (**E**) Venn diagrams show up-regulated genes H9-NSCs and iNSCs vs. HUCs. (**F**) Venn diagrams show down-regulated genes H9-NSCs and iNSCs vs. HUCs. (**G**) Gene ontology analysis of top 15 up-regulated genes in iNSCs vs. HUCs. (**H**) Gene ontology analysis of top 15 down-regulated genes in iNSCs vs. HUCs.

**Figure 5 cells-08-01043-f005:**
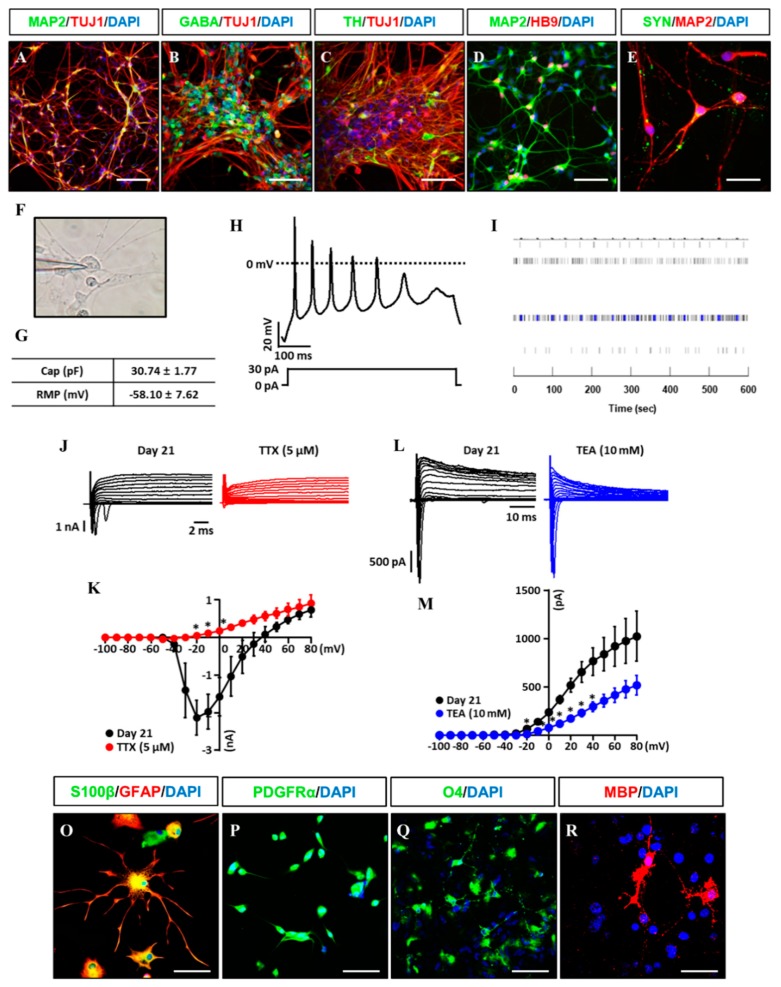
Differentiation potential of iNSCs in vitro. (**A**–**E**) Neuronal markers (TUJ1, MAP2, GABA, TH, HB9, and SYN) expression in iNSC-derived neurons, as determined by immunofluorescence analysis. (**F**) Image of patch clamp experiments. (**G**) Capacitance and resting membrane potential (RMP) were 30.74 pF ± 1.77 and −58.10 mV ± 7.62 in iNSC-derived neurons, respectively. (**H**) Representative trace of current clamp recordings of iNSC-derived neurons. Voltage responses were recorded from a holding potential of −80 mV using 500 ms steps with 30 pA of current injection. (**I**) MEA system showed spontaneous electronic spikes induced by iNSC-derived neurons. (**J**,**K**) The peak inward-currents due to Na+ channels were inhibited by addition of 5 μM Tetrodotoxin (TTX) (red, *n* = 3). (**L**,**M**) The outward-currents due to K^+^ channels were inhibited by addition of 10 mM TEA (blue, *n* = 4). For voltage-clamp recordings, sodium and potassium currents were recorded from a holding potential of −80 mV using 500 ms pulse voltage steps from −100 to + 80 in 10 mV increments. Values are mean ± SEM. * *p* < 0.05 vs. absence of TTX or TEA. (**O**) Astroglial markers (GFAP and S100β) expression in iNSC-derived astroglia, as determined by immunofluorescence analysis. (**P**–**R**) Oligodendroglial markers (PDGFRα, O4, and MBP) expression in iNSC-derived oligodendroglia, as determined by immunofluorescence analysis. Scale bars, 200 μm.

**Figure 6 cells-08-01043-f006:**
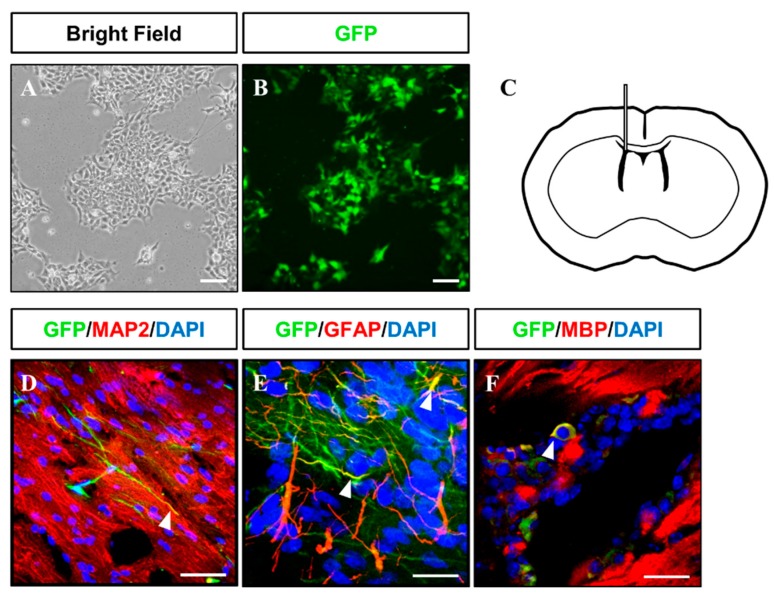
Injection of GFP-tagged iNSC into immunodeficient mouse brains. (**A**,**B**) Microscopy analysis of GFP-tagged iNSCs produced via lentiviral infection. (**C**) Schematic representation of iNSC transplantation. (**D**–**F**) At two months after transplantation, injected GFP+ iNSCs were positive for MAP2, GFAP, and MBP, as determined by immunofluorescence analysis. Scale bars, 200 μm.

**Figure 7 cells-08-01043-f007:**
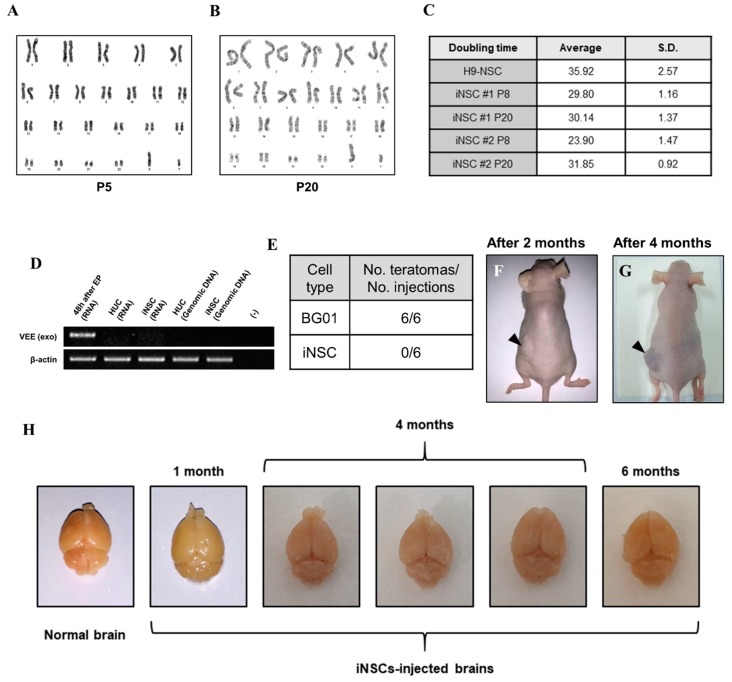
Biosafety evaluations of iNSCs. (**A**,**B**) Karyotyping of iNSCs at early (P5) and late (P20) passages. (**C**) Doubling time of iNSCs at early (P8) and late (P20) passages. (**D**) Exogenous Venezuelan equine encephalitis (VEEs) are not detected in total RNA and genomic DNA of iNSCs, as demonstrated by RT-PCR. (**E**) Teratoma formation efficiency of BG01-ESCs and iNSCs. (**F**,**G**) Teratoma formations in nude mice upon injecting 1 × 10^6^ BG01-ESCs in left dorsal flank and 1 × 10^6^ iNSCs in right dorsal flank. Black arrows indicate teratoma formations two months and four months after injection. (**H**) Morphological analysis of injected iNSCs into brain of nude mice. All mice did not die before sacrifice and no malignant appearances are showed in their brains at one month, four months, and six months after transplantation of iNSCs.

**Figure 8 cells-08-01043-f008:**
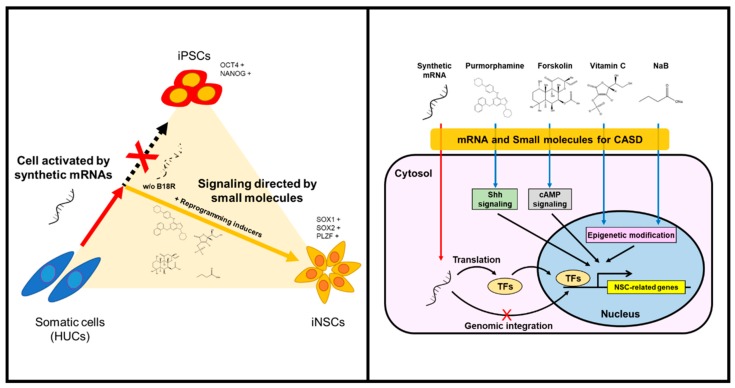
Schematic illustration of generating iNSCs from HUCs.

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
