# Peer review of "mRNA-Driven Generation of Transgene-Free Neural Stem Cells from Human Urine-Derived Cells"

_cells, 2019, doi:10.3390/cells8091043_

Round 1

Reviewer 1 Report

I commend the authors for the generation of the manuscript and their generation of transgene-free neural stem cells.  

However, the choice of reprogramming factors were typical of known attempts by other authors for example, ascorbic acid is well known and well considered however, why did the authors not consider other epigenetic modifiers that are currently commercially available since, this may have resulted in better outcomes?

Furthermore, it would have been good to examine the cell cycle of the iNSCs. Why was this approach not considered to analyse of the reprograming on the effect of cell cycle since this is crucial in some circumstances? Its well known that hypoxic conditions has detrimental effects on cell cycle?

Author Response

Reviewer #1

I commend the authors for the generation of the manuscript and their generation of transgene-free neural stem cells. 

However, the choice of reprogramming factors were typical of known attempts by other authors for example, ascorbic acid is well known and well considered however, why did the authors not consider other epigenetic modifiers that are currently commercially available since, this may have resulted in better outcomes?

→ We appreciate the reviewer’s excellent comment. To address this and make our rationale more clearly, because of an extremely low efficiency of reprogramming by the mRNA-based method itself, we expected performance of small molecules in the run-up to somatic reprogramming to iNSCs. In general, epigenetic modifiers regulate the epigenetic state within DNA and histone dynamics, therefore, we chose ascorbic acid as a DNA methylation modifier (Chen J et al., Nat Genet. 2013) and NaB as a histone acetylation modifier (Mali P et al., Stem Cells. 2010). As a result, we established an efficient and reproducible procedure for the generation of iNSCs.

As mentioned by the reviewer, other epigenetic modifiers could contribute to high-efficiency RNA-based reprogramming of somatic cells into iNSCs. However, the commercially available molecules including valproic acid, 5-azacytidine and RG108 provide the similar actions in their functions to ascorbic acid and NaB. In addition, the exposure of cells to many small molecules could induce cytotoxic effects. Accordingly, the protocol has been optimized in present. In agreement with your kind comment, we expect that further study on employing newly screened epigenetic modifiers could improve our protocol in a qualitative and quantitative manner. We added the explanation for this in the revised manuscript, page 18, lines 459-461.

Furthermore, it would have been good to examine the cell cycle of the iNSCs. Why was this approach not considered to analyse of the reprograming on the effect of cell cycle since this is crucial in some circumstances? Its well known that hypoxic conditions has detrimental effects on cell cycle?

→ It is a good point. Based on Ki67 staining (Figure 2K) and doubling time assessments (Figure 7C), the established iNSCs exhibited an active cell proliferation. Moreover, the reprogramming efficiency to iNSCs was enhanced under hypoxic conditions. Despite of the negative effects of hypoxia on cell cycle, many previous studies have emphasized the other effects of hypoxia on conversion to stem cells, including decrease in ROS damage, conversion in glycolytic metabolism and HIF induction, which elevates the reprogramming efficiency to NSCs and their proliferation capability (Yoshida Y et al., Cell Stem Cell. 2009; Saito S et al., Kaohsiung J Med Sci. 2015). We added the related statements in the revised manuscript, page 6, lines 251-252.

Reviewer 2 Report

The work by Kang and colleagues demonstrates the conversion of human urine-derived stem cells into iNSC by the use of a mRNA-replicative sytem in combination with pluripotency inducing small molecules. They showed the neuronal differentiation of the iNSC and some studies concerning safety issues of the cells in order to establish their clinical feasibility. The study describes an interesting approach, which might further approach towards clinical use, however, most result shown represent data from one patient only.

In general, most figures illustrate results from 1 donor. Is that representative? Please, comment and /or add data on reproducibility and significance. Please, include respective information to each figure legend (means from how many donors). Along this line, the presentation of results from different donors as shown in Fig. S5 reveals differences in cell density and cell morphology. Is the stability of expression of markers dependent on culture conditions (density, culture period, passage numbers)?

Line 270: What do the error bars mean? Is it the mean of several donors, or the mean of multiple cell cultures of one and the same donor? If the latter is the case, please comment on the reproducibility of induction efficacy.

Which passage numbers have been used for experiments shown in Fig. 7 D-G? How many donors? Show results from different donors.

Line 86: The harmless use of GLIS factors may not be without controversy. Please comment on the role and safety of GLIS factors, which have been reported recently to play a significant part in tumor control (e.g. Emerging Roles of GLI-Similar Krüppel-like Zinc Finger Transcription Factors in Leukemia and Other Cancers. By Anton M. Jetten DOI:https://doi.org/10.1016/j.trecan.2019.07.005

In Fig. 1 f, please correct x-axis legend (tretmant).

Author Response

Reviewer #2

The work by Kang and colleagues demonstrates the conversion of human urine-derived stem cells into iNSC by the use of a mRNA-replicative sytem in combination with pluripotency inducing small molecules. They showed the neuronal differentiation of the iNSC and some studies concerning safety issues of the cells in order to establish their clinical feasibility. The study describes an interesting approach, which might further approach towards clinical use, however, most result shown represent data from one patient only.

In general, most figures illustrate results from 1 donor. Is that representative? Please, comment and /or add data on reproducibility and significance. Please, include respective information to each figure legend (means from how many donors).

→ We appreciate the reviewer’s excellent comment. We used urine samples of 3 of male and 1 of female in this study. The analysis from sample of 1 of male (male, 33 ages) was presented as a representative for the establishment of iNSCs in the main manuscript. To demonstrate the repeatability and reproducibility of the present protocol, we provided the related results from the other samples (2 of male and 1 of female) in Supplemental Figure 2, 3 and 5. As the reviewer suggested, we added detailed descriptions about the samples in the revised manuscript, Materials and Methods, page 3, lines 98-100.

Along this line, the presentation of results from different donors as shown in Fig. S5 reveals differences in cell density and cell morphology. Is the stability of expression of markers dependent on culture conditions (density, culture period, passage numbers)?

→ We appreciate the reviewer’s comment. In general, the established iNSCs were subcultured at 1:3~1:4 ratios under the standard culture condition previously described (Wenlin Li et al., Proc Natl Acad Sci U S A. 2011). In this environment, iNSCs stably expressed NSC markers while actively proliferating. We used 5~10 passage of iNSCs for evaluation. As the reviewer suggested, we added detailed descriptions for the samples in the revised manuscript, Materials and Methods, page 3, lines 118.

Line 270: What do the error bars mean? Is it the mean of several donors, or the mean of multiple cell cultures of one and the same donor? If the latter is the case, please comment on the reproducibility of induction efficacy.

→ Thanks for your comment. In Figure 1, the error bars meant multiple cultures by using the cells derived from the same donor. We first optimized the condition for establishing iNSCs, applied to samples of the other donors, and successfully generated iNSCs. We understand the reviewer’s concerns, however, there were the reasons why we carried out the experiments in this manner. HUCs derived from human urine are primary tissue-derived cell line, indicating heterogeneous cell populations, in contrast to other commercial cell sources (Dörrenhaus A et al., Arch Toxicol. 2000). Moreover, this variation largely depends on individual donors. Therefore, please understand that it is very difficult to provide absolute values in the induction efficiency.

Which passage numbers have been used for experiments shown in Fig. 7 D-G? How many donors? Show results from different donors.

→ We used 5-10 passage of iNSCs for analyzing. As suggested, we added detailed descriptions for the samples in the revised manuscript, Materials and Methods, page 3, lines 118.

Line 86: The harmless use of GLIS factors may not be without controversy. Please comment on the role and safety of GLIS factors, which have been reported recently to play a significant part in tumor control (e.g. Emerging Roles of GLI-Similar Krüppel-like Zinc Finger Transcription Factors in Leukemia and Other Cancers. By Anton M. Jetten DOI:https://doi.org/10.1016/j.trecan.2019.07.005

→ We appreciate the reviewer’s excellent comment. To our knowledge, most of reprogramming factors are tumorigenic. Therefore, continuous expression of exogenous reprogramming factors can induce tumorigenesis. To overcome the problem, we suggested the combination of controllable mRNA delivery system, allowing mRNAs to be transiently expressed, and small molecules for non-genomic integration to the cells.

In Fig. 1 f, please correct x-axis legend (tretmant).

It is an obvious mistake. This was corrected in the revised version and we double-checked throughout the manuscript.

Round 2

Reviewer 2 Report

The issues raised by this reviewer have been largely addressed satisfactorily.